# Cancer Therapy Challenge: It Is Time to Look in the “St. Patrick’s Well” of the Nature

**DOI:** 10.3390/ijms221910380

**Published:** 2021-09-26

**Authors:** Gregorio Bonsignore, Mauro Patrone, Federica Grosso, Simona Martinotti, Elia Ranzato

**Affiliations:** 1DiSIT–Dipartimento di Scienze e Innovazione Tecnologica, University of Piemonte Orientale, Viale Teresa Michel 11, 15121 Alessandria, Italy; gregorio.bonsignore@uniupo.it (G.B.); mauro.patrone@uniupo.it (M.P.); simona.martinotti@uniupo.it (S.M.); 2Mesothelioma and Rare Cancer Unit, Azienda Ospedaliera SS Antonio e Biagio e Cesare Arrigo, 15121 Alessandria, Italy; fgrosso@ospedale.al.it; 3Translational Medicine Unit, Dipartimento Attività Integrate Ricerca e Innovazione (DAIRI), Azienda Ospedaliera SS Antonio e Biagio e Cesare Arrigo, 15121 Alessandria, Italy; 4DiSIT–Dipartimento di Scienze e Innovazione Tecnologica, University of Piemonte Orientale, Piazza Sant’Eusebio 5, 13100 Vercelli, Italy

**Keywords:** ascorbic acid, cancer, cancer therapy, capsaicin, curcumin, epigallocatechin-3-gallate, natural compounds, resveratrol, synergy

## Abstract

Cancer still remains a leading cause of death despite improvements in diagnosis, drug discovery and therapy approach. Therefore, there is a strong need to improve methodologies as well as to increase the number of approaches available. Natural compounds of different origins (i.e., from fungi, plants, microbes, etc.) represent an interesting approach for fighting cancer. In particular, synergistic strategies may represent an intriguing approach, combining natural compounds with classic chemotherapeutic drugs to increase therapeutic efficacy and lower the required drug concentrations. In this review, we focus primarily on those natural compounds utilized in synergistic approached to treating cancer, with particular attention to those compounds that have gained the most research interest.

## 1. Introduction

Cancer is considered, in our century a leading cause of death and the single most significant obstacle to increasing life expectancy in every country in the world.

So, cancer remains a worldwide challenge with significant influence, not only on human health, but also on the global economy.

According to World Health Organization (WHO) data, cancer is reported to be the first- or second-most common cause of death in people under 70 years of age in 91 of 172 countries, and the third- or fourth-most common in an additional 22 countries [1]. Neoplasm incidence and mortality are fast rising all around the world, reflecting both the population’s increasing growth and age.

In addition, the escalating rise of tumours as a cause of death is nearly equal to the noticeably declining mortality rates for coronary heart disease and stroke in many countries. It is interesting, today, to notice that changes in tumour incidence are most pronounced in emerging economies. There, history is repeating itself in the shift from poverty- and infection-related malignancies to those diseases that are already dominant in developed areas (e.g., in North America and Europe). These kinds of cancers are often described as “caused by the westernization of lifestyle”, however, the differing cancer profiles in several countries and regions show that marked geographic disparities still exist; in fact, there are local risk factors persisting in populations at quite different phases of social and economic transition.

The significant changes in infection-associated cancer rates (liver, cervix, and stomach), described in countries with different levels of economic development, confirm this theory [1].

Even if substantial improvements have been achieved in early diagnosis and in new drug improvement, there is a strong need to develop new methodologies and new molecules that are able to advance the therapeutic approaches currently available [1].

## 2. Natural Compounds for Cancer Therapy

In ancient times, as reported in the medical literature, doctors used both surgery and natural compounds (especially plant products) to treat patients. There is, similarly, historical evidence for the use of natural compounds in traditional Chinese medicine and Indian Ayurvedic practise.

Natural compounds from the plant, microbial and fungus kingdoms represent an uncountable resource of new molecules, potentially usable as antitumor remedies if their availabilities, toxicities and activities are tolerable.

There is much data supporting the use of natural compounds in cancer treatment; nevertheless, the validity of their use is not completely verified by scientific evidence.

Natural compounds represent an interesting point of comparison with current health culture. Natural products are an important option in cancer therapy today; there are currently a significant number of anticancer agents available, both natural and derived from natural products (from animals, plants, and microorganisms, also from marine environment) [2].

Recently, natural product-based drug discovery has been growing due to the development of new approaches, such as combinatorial synthesis and its associated methods.

There are many examples of plant-derived compounds; for example, irinotecan, vincristine, paclitaxel and etoposide. Microbes are also proving to be an important source thereof; mitomycin C, actinomycin D, bleomycin, l-asparaginase and doxorubicin are drugs obtained from bacteria. In addition, citarabine is the first drug originating from a marine source [3].

Today, a new generation of taxanes, anthracyclines, alkaloids from Vinca, camptothecins and epothilones have been developed. Some of these are already in clinical use, others are under study.

Other molecules deriving from marine animals and plants (e.g., trabectedin, ET-743, bryostatin-1, neovastat) have also reached clinical trials.

There are many different classes of natural compounds, such as terpenes, carotenoids, phenolic compounds (flavonoids, stilbenes, phenolic acids, tannins, coumarins), alkaloids, nitrogen compounds and organosulphates (isothiocyanates and indoles, allysulphates), which have aroused much interest [3].

Marine environments have yielded several classes of compounds with diverse biological activities [4]. Many molecules extracted from marine organisms have been investigated recently; for example, arabinosides (isolated from marine sponge *Cryptotethya crypta*), didemin B (*Trididemnum solidum*) and bryostatins (*Bugula Neritina* and other Bryozoa) have shown anticancer activity [5].

Soil fungi contain a huge number of defence metabolites, allowing them to survive amidst other organisms (other fungi, nematodes, insects and bacteria) and helping them to inhabit more preferential areas by way of effective antagonism. Non-cytotoxic and host-mediated antitumor polysaccharides have been obtained from various fungi (Basidiomycetes and Ascomycetes) [6].

Many organisms produce venoms and toxic substances; these compounds are very attractive candidates for drug development. They often show physiological effects on animals, humans among them. Examples include cantharidin, which is produced by blister beetles, or solenopsins, produced by fire ants (*Solenopsins invicta*), among others [6].

These drugs from natural environments exert their effects on cancer cells by a plethora of mechanisms of action, for example, interference with tumor signal transduction, the inhibition of topoisomerases I or II, DNA alkylation, interaction with microtubules, etc. [7].

Several natural molecules are able to interact with signalling pathways and regulate the gene expression involved in cell differentiation, cell cycle regulation, and apoptosis [8].

## 3. Synergy

Cancer is a multifaceted clinical condition in which several cellular and molecular pathways are altered. Each cancer type expresses a specific genetic fingerprint.

However, all cancers share a mutual behaviour centred on unrestrained proliferation, migration, and invasion. This aggressive phenotype is the tangible clinical problem and remains unsolved.

Recent anti-tumour strategies, as well as drug discovery efforts, are constantly growing the therapeutic resource base, with important improvements being made in terms of personalised options [9].

An important number of drugs, such as antibody-based drugs or particular inhibitors of specific targets, have been developed following the concept “one-drug-one-target” [10].

Yet, the heterogeneity of some molecules, such as those from natural origins [11], permits their potentially multi-targeted affect.

In fact, these compounds can reach several cellular and molecular targets, altering some pathways or diverse signalling cascade elements.

Furthermore, this multi-targeted activity is not only due to single compound efficacy, but also to a combination of molecules, as present in numerous natural extracts. So, each compound may be able to reach one or more targets, increasing the pharmacological activity of the whole extract [12].

Another intriguing option is the combination of natural extracts, or at least of their main components, with conventional chemotherapy, with the main aim of reducing the incidence of resistance, thereby increasing cancer cell toxicity and generally diminishing the injuriousness of chemotherapy drugs [13].

Today, growing evidence has pointed out that combined therapies are much more effective than single-drug-based treatments. Therefore, a combination of different therapies is deployed to treat not only neoplasms but also other illnesses, such as viral and bacterial infections, inflammatory diseases, etc. [14].

Combined therapies normally require the co-administration of two or more drugs. These combinations may be combinations of pure compounds or drugs based on mixtures from natural extracts.

Synergy is consequently the most appropriate characteristic of combined therapies, including those using natural extracts.

In pharmacology, the sum synergic effecta of some combinations are more powerful than their individual components, separately. Synergy is not a binary factor; the pharmacological interaction between the components of a mixture can be synergic to varying degrees.

Combinations of natural products with cancer drugs that exhibit synergy have been developed to further improve upon existing treatment strategies, due to the propensity of some natural products to provide improved therapeutic efficacy or overcome drug resistance with decreased risk for adverse side effects and toxicity in normal cells [12,15].

Specific natural compounds and constituents have been extensively demonstrated to lessen chemotherapy drugs’ cytotoxic activity in various cell lines, thereby widening the therapeutic window and also lowering required drug concentrations, while providing the same effect [13].

In Table 1, we summarize a list of natural molecules with their molecular effects on cancer cells and their synergistic effects in combination with chemotherapeutic agents in some cancer models.

## 4. Natural Compounds and Sinergy

After the extensive review of many natural compounds of different origins, in the present work we want to focus, in particular, on those compounds that, over the years, have collected the greatest interest from researchers, demonstrating synergistic behaviours with some drugs used in the therapies of different tumors (Table 2).

### 4.1. Ascorbic Acid

Ascorbate is an important redox cofactor and catalyst for many biochemical reactions. In humans, it cures or prevents scurvy (this word comes from *skjoerberg* or *skorbjugg* that, in the Scandinavian language, mean ‘rough skin’) [98,99,100].

Vitamin C is contained in several well-known plants and fruits, but also in animal organs (brain, kidney, liver), yeasts and prokaryotes (but not in cyanobacteria) [101]. In superior plants, vitamin C is obtained from D-glucose, and it is involved in many metabolic processes, for example the scavenging of H_2_O_2_ [102], the maintenance of the α-tocopherol pool [103] and, also, it behaves like violaxanthin deoxidase cofactor [104].

In animals, ascorbic acid is synthetized from glucose by enzyme L-gluconolactone oxidase, and has been found in the kidneys of reptiles but, in mammals, it is found in the liver. Most animals can synthesize ascorbic acid, but some species (humans, other primates, guinea pigs, Passeriformes birds and flying mammals) must get it from their diets. This condition is due to a deficiency in gluconolactone oxidase that does not allow the conversion of L-gluconolactone to 2-keto-L-gluconolactone [105].

Vitamin C, as highlighted in *Vitamin C and the Common Cold* [106], is linked to the immune system. This is confirmed by the rapid decrease in ascorbate and leucocytes during stress and infection [106]. Ascorbate is also an antiviral agent; it acts against viruses aiding in the degradation of their nucleic acids [107,108,109]. Furthermore, Vitamin C accelerates the destruction of histamine, a molecule that mediates allergy and cold symptoms, reducing it by 30–40% [110,111,112].

Recently, Vitamin C was studied on some chronic diseases, in particular, atherosclerosis. An ascorbate deficiency (<0.2 mg/dL) is inversely related to this condition [113,114]. Atherosclerosis is related to the oxidation of low-density lipoprotein, but the consumption of Vitamin C by hospitalized patients can reduce myocardial infarction [115] and, in acute smokers, which have two-fold LDL oxidation levels, ascorbate supplementation can reverse LDL peroxidation [116,117,118].

The poor intake of Vitamin C and E is related to an increase in fractures, especially in female smokers, up to five-fold. However, among women smokers with a high intake of both vitamins (>200 mg/day), the probability of fractures is not increased [119].

The use of ascorbate as a cancer therapy is under controversy. Seventy years ago William McCormick [120] described how tumour patients often showed low blood levels of vitamin C and featured scurvy-like symptoms, leading him to assume that vitamin C might protect against cancer by increasing collagen synthesis. In 1972, Ewan Cameron theorised that ascorbate could suppress cancer development by inhibiting hyaluronidase, weakening the extracellular matrix and enabling tumours to form metastases. In 1976, Cameron and Pauling published a study of 100 patients with terminal neoplasms treated with ascorbate. Even though the study was not guided by modern clinical standards, mainly because they lacked a placebo control group, their results revealed that ascorbate-treated patients showed improved quality of life of life and increased mean survival time [121,122]. Other clinical trials have independently indicated similar results. So, interest in the potential of ascorbate for tumour treatment grew.

However, double-blind randomized clinical trials directed by Charles Moertel of the Mayo Clinic failed to show any positive effects of high-dose vitamin C in cancer patients [123]. So, the enthusiasm for the results obtained by the Cameron–Pauling trials was dampened by these data and the research on ascorbate was silenced for many years.

At the beginning of the 2000s, some studies at the National Institute of Health (NIH) established dietary recommendations for ascorbate [124,125]. When people received oral doses, low plasma concentrations of ascorbate were achieved (around 100–200 μM), while intravenous administration allowed 100-fold higher concentration than oral (around 15 mM) [126]. This is due to partial intestinal absorption, excretion and renal re-absorption during oral administration. Intravenous administration evades this control, allowing high plasmatic concentrations [126]. So, a high (“pharmacologic”) ascorbate level is achievable only with intravenous administration, not with oral administration (“physiologic” level). So, while only pharmacologic vitamin C level could be considered as a drug, the attention for using ascorbate as an anti-tumour agent has re-emerged.

After the basic information about ascorbate pharmacokinetics was understood, some studies described the effects of ascorbate on cancer cells. The in vitro analyses showed that ascorbic acid, at around 20 mM concentration, is able to selectively kill cancer cells, without affecting normal cell lines [127]. Additionally, other authors found that ascorbate toxicity in cancer cells was due to hydrogen peroxide formation, with ascorbate radical as an intermediate [128,129].

Some studies also explored the intravenous administration of vitamin C in cancer patients. Padayatty and co-workers and Hoffer and colleagues demonstrated that intravenous ascorbate, at high doses, is well tolerated by patients with different cancer types [126,130]. Other studies have highlighted that ascorbate. administered intravenously, improves quality of life of life in cancer patients [131].

Some studies concentrated their attention on the intravenous effect of ascorbate in cancer patient’s survival. Ascorbate treatment could increase quality of life and decrease chemotherapy-related side effects in cancer patients [130,132].

Moreover, in vitro tests [15,133] and in vivo xenografts have also demonstrated the synergistic effects of ascorbate with other classic anti-cancer drugs [16]. Recently, some clinical studies are demonstrating that ascorbate, in combination with chemotherapeutic drugs, displays promising clinical efficacy [134,135].

### 4.2. Curcumin

Curcumin has fascinated humankind since ancient times due to its numerous biological effects, including anti-antioxidant, inflammatory and antitumor abilities [136]. Curcumin is obtained from the rhizome of *Curcuma longa* (ginger family) and recognised, from a chemical point of view, as 1,7-bis-(4-hydroxy-3-methoxyphenyl)-hepta-1,6-diene-3,5-dione. *Curcuma longa* extracts and its components have been utilized in traditional Chinese medicine for thousands of years.

A huge number of experiments, both in vitro and in vivo, have established that curcumin could impede various cancers’ growth including ovarian, gastric and colorectal neoplasms, by activating apoptosis.

Although it is well tolerated by patients, curcumin is poorly adsorbed by organisms, therefore, it is very difficult to utilize. Some approaches have tried to improve its bioavailability; in particular, the main strategies are its combination with adjuvants, the utilization of chemical analogues and the development of novel delivery approaches.

The pharmacodynamics data available for humans are limited and there is debate as to whether its efficacy is due to some curcumin components, or to other mechanisms, acting indirectly. At present, several clinical trials (phase I or II) are ongoing to investigate the benefits of curcumin as a chemo-preventive and chemotherapeutic agent in a variety of tumours [137,138,139].

A clinical study (phase I) on twenty-five patients with different lesions both (pre-malignant and high-risk lesions) revealed that oral curcumin can be chemopreventive [140,141]. The Cleveland Clinic carried out research with five patients with familial adenomatous polyposis, who were treated three times a day with a combination of curcumin and quercetin for a mean duration of 6 months. Data showed that polyps’ number and size were decreased in all patients, compared with controls [142].

Carrol et al. [143], in a recent phase II clinical trial, investigated curcumin and its potential activity for prevention of colorectal neoplasia in smokers with aberrant crypt foci (ACF). The results showed a significant reduction of ACF number by a 4-g dose curcumin at the level and indicated that curcumin may have cancer-preventive effects against pre-invasive neoplastic lesions [143].

Unfortunately, interpretation of this study is limited, asACF is a controversial biomarker of colon carcinogenesis.

Human pancreatic cancer treatment with curcumin has been evaluated in a phase II clinical trial. Curcumin (8 g) was administered to 25 patients orally, daily, with restaging every two months, of whom 21 were evaluable for response [139].

Some phase II clinical studies suggested that a gemcitabine and curcumin combination is a conceivable treatment for pancreatic cancer patients [144].

In a study by Bayet-Robert et al. [145], 14 advanced and metastatic breast cancer patients were treated with curcumin and docetaxel combined therapy. The research confirmed that this combination reduced the level of vascular endothelial grow factor (VEGF) with optimistic results [146].

### 4.3. Epigallocatechin-3-gallate

Tea is a popular diffusion beverage, obtained from the plant *Camellia sinensis*. There are many chemical compounds in tea, but the most abundant are catechins (especially in green tea), which include: (-)-epigallocatechin-3-gallate (EGCG), (-)-epigallocatechin, (-)-epicatechin-3-gallate and (-)-epicatechin.

EGCG represents more than 50% of the total catechins, is one of the best-studied constituents of green tea and appears to be the most effective one. EGCG seems promising for chemoprevention according to in vitro, animal, clinical and epidemiological studies. EGCG can induce the reduction of some cancer cell lines’ growth and apoptosis in vitro [127,147], inhibiting tumour incidence in in vivo experiments, for example colon, liver, lung, mammary glands, pancreas, prostate, and skin neoplasm models [148].

Anticancer effects ascribed to EGCG are: antioxidant activities, apoptosis induction, carcinogen metabolism modification, cell cycle arrest, DNA damage prevention, metastasis inhibition modulation of multiple pathways of signal transduction and proteasome blocking [149].

Yoshizawa et al. in 1987, described how EGCG administration suppressed 7,12-dimethylbenz[a]anthracene (DMBA) plus teleocidin-initiated carcinogenesis. EGCG causes a significant reduction in the incidence of tumours compared with controls [150].

However, the EGCG antitumoral effect observed in animals is not confirmed, at moment, for green tea consumption in humans. Probably, these epidemiological studies’ inconsistent results were caused by various confounding factors, for example the quantity and the quality of the tea consumed or, possibly, the effect of caffeine [151].

Moreover, EGCG tissue and plasmatic concentrations obtained by the oral intake of tea are lower than the effective concentrations utilized in in vitro experiments (10–100 μmol/L). To avoid these kinds of problems, better-designed clinical studies have been designed; for example, EGCG-enriched fractions such as polyphenon E, a well-defined green tea catechin (GTC) extract, or highly purified EGCG, which have been provided by pharmaceutical companies [148].

Systemic bioavailability analyses in human volunteers of orally administered catechins have been already performed. Chow et al. examined the tolerability, safety, and pharmacokinetics of EGCG and polyphenon E, administered at doses ranging from 200 to 800 mg [152,153].

Some recent trials have confirmed EGCG’s chemopreventitive and chemotherapeutic role, offering more details on its action in the human body. For example, Ahn et al. described that oral treatment with polyphenon E or purified EGCG (200 mg daily for 3 months) was effective in patients with cervical lesions infected by human papilloma virus (HPV) [154].

In Japan, green tea extract’s (GTE) effect on metachronous colorectal adenomas was evaluated. GTE, administered orally (1.5 g/d for 12 months) in addition to a tea-drinking lifestyle, has been shown to be useful in reducing the incidence of metachronous adenoma in people 1-year post-polypectomy.

Another application of catechins is the chemoprevention of prostate cancer (oral GTCs). Sixty volunteers with high-grade prostate neoplasia received either 600 mg of GTCs or a placebo, daily, for 1 year. After 1 year of follow-up, only 3% of patients that had received oral GTCs showed a prostate tumour, while 30% of patients in the placebo group developed cancer. According to these observations, treatment with GTCs was able to reduce prostate cancer diagnoses by almost 80% [155].

From several clinical studies it is emerging that EGCG is an active cancer suppressor, with limited side effects and high safety. In addition, it has shown synergistic effects with other anticancer drugs like chrysin, curcumin, erlotinib, etoposide, 5-fluorouracil, tamoxifen and temozolomide [156,157]. However, it can also inhibit some anticancer treatments (bortezomib and other proteasome inhibitors) [158,159].

### 4.4. Resveratrol

Resveratrol is a polyphenol, isolated for the first time in 1940. It is an ingredient of white hellebore roots (*Veratrum grandiflorum* O.Loes) contained in various food sources including grapes, mulberries, red wine and peanuts. In 1963, resveratrol was recognised as the active element in *Polygonum cuspidatum* roots, a plant used in Japanese and Chinese traditional medicine.

It can inhibit tumorigenesis at multiple phases, including initiation, as well as during a cancer’s progression [160]. Several other studies confirmed the strong chemo-preventive resveratrol efficacy in in vivo carcinogenesis models. In mice and rats, the oral or local application of resveratrol reduced DMBA-initiated and 12-otetradecanoylphorbol-13-acetate-promoted skin cancers, repressed DMBA-induced mammary carcinogenesis, inhibited 1,2-dimethylhydrazine-induced carcinogenesis of the colon epithelium and N-nitrosomethylbenzylamine-induced esophageal tumors [161].

Moreover, extensive study has suggested that resveratrol might be an important candidate for cancer therapy because it could act by interfering with many signalling pathways playing pivotal roles for cell growth, cell death, inflammatory process, angiogenic mechanisms and metastasis formation.

Besides, resveratrol was also described to show exhibit synergistic effects with other classic anti-cancer drugs, such as doxorubicin, cisplatin and vinorelbine.

The first clinical trial of resveratrol in colon cancer patients was performed with the aim to assess low dose effects of a plant-derived resveratrol formulation and resveratrol-containing freeze-dried grape powder (GP). Treatment was administrated to 8 patients received for 14 days until the day prior to surgery for colon cancer resection.

The two compounds showed significant ability to inhibiting Wnt pathway targets on normal colon mucosa, whereas GP treatment augmented some Wnt target genes expression in colon cancer. Therefore, resveratrol may show more clinical utility as colon cancer prevention agent rather than for established colon cancer treatment [162,163].

A randomised, double-blind, phase I, clinical trial, showed the SRT501 (micronized resveratrol) effects in colorectal cancer and hepatic metastases patients. In malignant hepatic tissue following SRT501 treatment, a marker of apoptosis, i.e., cleaved caspase-3, was significantly increased by 39% compared to tissue from placebo-treated patients [164].

Moreover, a recent study suggested that resveratrol could attenuate the paclitaxel’s anticancer efficacy in some breast cancer cell lines and in vivo experiments [165], but it has also shown synergistic effects with other anticancer drugs like cisplatin, doxorubicin and vinorelbine [166,167].

### 4.5. Capsaicin

Capsaicin (trans-8-methyl-N-vanillyl-6-nonenamide) is a homovanillic acid derivative and represents the major spicy component in red and chili peppers.

Capsaicin has been studied in the past for medical applications such as in anti-oxidant, anti-inflammatory and analgesic compounds [24].

Recently, some researchers have explored the benefits of capsaicin as an anti-cancer agent, with a detailed analysis of the molecular mechanisms induced by its exposure. In fact, capsaicin is able to influence the expression of some genes, in different types of tumor models, that are directly involved in cell growth, apoptosis, metastatization and angiogenesis processes [168].

The interest in capsaicin is also due to its possible combinational use with other chemotherapeutics drugs or dietary molecules, highlighting its synergistic antitumor activities.

Capsaicin combined with resveratrol was able to induce apoptotic pathways by nitric oxide (NO) elevation in a p53-dependent way [25].

Capsaicin has shown an interesting synergistic behavior in association with genistein, acting in breast tumor cell lines through AMPK and cyclooxygenase 2 regulation [169].

Moreover, capsaicin and brassinin, an indole obtained from cruciferous vegetables, possess synergistic antitumor abilities in regulating matrix metalloproteinases, thereby reducing migration and the invasion of prostate carcinoma cell lines [170].

## 5. Natural Compounds as Epigenetic Modulators

An increasing number of scientific reports highlight the implication of genetic and epigenetic alterations that can lead to the alteration of transcription factors, oncogenes’ over-expression, tumor suppressor genes’ inactivation, producing a deregulation of signaling pathways, and, finally, tumor occurrence [170,171].

The term “epigenetics” has been utilized to include the heritable changes in DNA and protein alterations that lead to a disturbed expression of genes involved in cell growth and cell cycle progression, cell death, metabolism, etc. [172].

Epigenetic alterations are hypothetically reversible; so, they are interesting for the development of new anti-tumor strategies [173].

Drugs able to target epigenetic mechanisms could represent the frontier of a new chemotherapeutic approach, and natural compounds have demonstrated their great potential [174].

It has been demonstrated that a vegetables- and fruits-rich diet can significantly reduce the risk of tumor growth. This is mainly due to the presence of some phytochemicals that are able to modulate oncogenes’ expression and tumor suppressor genes [175].

In fact, some natural compounds have been described as being able to influence some of the epigenetic processes involved in carcinogenesis, such as the modification of histone proteins (acetylation and methylation), DNA methylation and microRNA expression [176].

The natural compounds most studied in epigenetic processes in tumorigenesis are EGCG, curcumin and resveratrol. In particular, EGCG could epigenetically reactivate p21/waf1, Bax and PUMA in prostate cancer cell lines, promoting the block of cell cycle and cell death induced by degradation at the proteasome of histone deacetylases (HDACs) [177]. EGCG is also able to repress the androgen receptor (AR) hormone response by the reduction of AR acetylation. This phenomenon determines a reduction of prostate cancer cells’ growth, promoting apoptosis [178]. Moreover, EGCG has been also described as a potential epigenetic modifier of HDACs, restoring epigenetically silenced genes in cervical and skin tumors. EGCG could also reactivate the WIF (Wnt inhibitory factor-1) expression by demethylation of the gene promoter, inducing cell growth arrest and influencing the Wnt pathway in A549 and H460 lung tumor cell lines [179].

Moreover, EGCG reactivates the expression of WIF-1 (Wnt inhibitory factor-1) through promoter demethylation and inhibits cell growth by downregulating the Wnt canonical pathway in H460 and A549 lung cancer cell lines [171].

EGCG sensitizes ERα-negative cancer cells to respond to 17β-estradiol, and the antagonist tamoxifen. EGCG associated with trichostatin A (TSA, a HDAC inhibitor) reactivates the ERα expression in MDA-MB213 cells (a triple-negative breast cell line) by influencing histone methylation and acetylation, thus remodeling chromatin assembly [180].

Curcumin has been evaluated as an excellent non-toxic hypo-methylating molecule for breast cancer therapeutic approaches [181]. For example, curcumin influences astrocyte differentiation, promoting neural differentiation-inducing histone (H3 and H4) hypo-acetylation [175].

Resveratrol induces, in p53-wild type and p53-mutant prostate tumor cells, the downregulation of metastasis-associated protein 1 (MTA1), promoting the destabilization of its nucleosome, remodeling deacetylation co-repressor complex. This complex is able to mediate the histone and non-histone post-translational modifications inducing transcriptional repression [176].

Taking these data together, natural compounds express a real potential effect for cancer therapies due to their reverting effects on epigenetic modifications in the oncogenes and tumor-suppressor genes involved in cancer’s development and growth.

## 6. Conclusions

In the current century, cancer appears to be the most challenging pathology to treat; therefore, new, well-tolerated and effective therapeutic approaches are necessary.

The primary problem in cancer therapy is drug resistance; a huge number of cellular mechanisms are involved in this resistance [172] and no molecule is excluded from acquiring a resistant phenotype. In this regard, the risk of resistance could be minimized by combined therapies; if tumour cells gain resistance against one of the drugs, other components of the mixture can still have an impact on them.

Even though natural compounds and their extracts, sometimes, can exhibit resistance phenomena, pure natural compounds can be utilized in combination with another agent to reduce the development of resistance, but natural extracts are, themselves, a blend, acting as a synergic therapy, and contributing to the reduction of drug-resistant phenotypes [173].

Natural extracts may share some disadvantages with classical cancer drugs. Resistance is not the only problem, in fact, poor bioavailability is a common problem due to their very different structures. This results in poor absorption, high metabolism rates and a rapid excretion process. In all these cases, low plasma concentrations are reached. On the other hand, some natural compounds are absorbed rapidly and completely, entering the plasma in their native form, thus reaching significant plasma concentrations.

Therefore, the poor bioavailability of some natural extracts really hinders these natural product’s potential to be developed into a clinically approved drug. In fact, this poor bioavailability requires a long-term dosing strategy. Another problem is due to the wide patient population required, representing significant drug exposure variability for natural extracts. So, bioavailability represents an important concern for the possible use of natural compounds as drug candidates.

Therefore, a new strategy is these of drug delivery systems to precisely target given body parts. This option might solve these critical issues [174]. Nanotechnology could play a noteworthy role for advanced drug preparations, controlling both drug release and delivery. The green chemistry-design approach of for the loading of nanoparticles with drugs can also be very useful in minimizing the hazardous constituents of the biosynthetic process. Thus, these green drug-delivery nanoparticles could reduce the side-effects of medications [175]. However, drug’s precise release at determined sites, assessments of their effects at the cellular and tissue levels, and the required predictive mathematical modelling have not yet been developed [176].

Moreover, another interesting application of natural compounds is that the onset of resistance is made more difficult by the use of natural extract, thanks to their poly-pharmacological properties and, as has happened with common drugs, bioavailability problems can be solved with novel approaches, such as encapsulation [177], nanoparticles [178], liposomes [179] or emulsions [180]. These approaches ameliorate the bioavailability of hydrophilic compounds with poor absorption or low stability and increase the solubility of highly hydrophobic compounds and extracts [172,181].

The issues of reproducibility and the complexity of natural mixtures are the most important drawbacks thereof. Furthermore, positive in vitro data are not directly correlated to positive in vivo data due to poor solubility and, consequently, lesser accumulation at the target site, leading to a significant increase in systemic toxicity.

In conclusion, it will be crucial to understand the signaling pathways involved, as well as bioavailability and true cytotoxicity of these natural products, in answering the growing demand for the evaluation of natural products in clinical trials.

## Figures and Tables

**Table 1 ijms-22-10380-t001:** Natural products, their effects on cancer cells and their synergy with chemotherapeutics.

Natural Product	Common Sources	Mechanisms of Action	Chemotherapy Drug and Synergistic Effects
Ascorbate	//	Ascorbate’s toxicity on cancer cells is due to hydrogen peroxide formation with ascorbate radical as an intermediate state [15].	Ascorbate showed synergistic effects in in vitro [15] and in vivo experiments with common anti-cancer drugs [16]. Ascorbate is used in synergistic approaches at mM concentrations.
Podophyllotoxin	*Podophyllum*	Podophyllotoxin inhibits tubulin polymerisation, arresting the cell cycle at metaphase [17]. It is effective in treating Wilms’ tumours, various types of genital cancer and in non-Hodgkin’s and other lymphomas [18], lung tumours [19,20], and neuroblastoma [21].	Podophyllotoxin (ranging from 7.5 to 15 nM) showed synergistic effects with [21]: - Cisplatin - Methotrexate
Neem extract	*Azadirachta indica*	Neem components’ modify the tumour environment, decreasing vessel formation [22]. They are employed against cervical, breast, and ovarian cancer.	Neem extract showed synergistic effects with [23]: - Cisplatin - Genduin
Capsaicin	Red and chili peppers	The antitumor mechanism of capsaicin increases apoptosis and cell cycle arrest [24]	Capsaicin (at µM concentrations) showed synergistic activities with other agents, such as resveratrol and genistein [25].
Curcumin	*Curcuma longa*	In colon cancer, AMPK causes invasion through the inhibition of NF-*k*B, uPA, and MMP9. Curcumin is able to inhibit this pathway [26]. In fact, it can reduce TNF-a, COX-2, and IL-6 production, contrasting inflammation [27]. It also inhibits cell proliferation by increasing the activity of glutathione-S-transferases and p21, which are, respectively, a biotransformation enzyme and a cell-cycle protein [28,29]. Moreover, it increases some pro-apoptotic proteins’ expression (Bax, Bim, Bak, Noxa) while inhibiting anti-apoptotic elements (Bcl-2, Bcl-xL) [30]. In addition, curcumin reduces the expression of VEGF and matrix metalloproteases, preventing metastases’ development [31].	Curcumin (at µM concentrations) shown synergistic activity with: - Bevacizumab [32] - Capecitabine [33] - Dasatinib [34] - FOLFOX [34]
Resveratrol	Almost 70 plant species	Resveratrol interacts with several targets. In fact, it acts on cytochrome P450 isoenzymes and may downregulate some elements that are often upregulated in tumoral cells. Among these are cyclooxygenase inflammation mediator enzymes and NF-kB transcription factor [35].	Resveratrol (for clinical studies, at concentrations ranging from 20 to 120 g/day) showed synergistic effects with [36]: - 5-FU - Etoposide - Mitomycin - Oxaliplatin - Curcumin
GLC (*Ganoderma lucidum extract)*	*Ganoderma lucidum*	GLC induces NK cell toxicity by NKG2D augmented expression and natural cytotoxicity receptors (NCR), increased intracellular MAPK phosphorylation, and the secretion of granulysin and perforin [37]. In some human cell lines, *Ganoderma lucidum* promotes the arrest of mitosis [38].	
GLP (Polysaccharides extracted from Ganoderma lucidum)	//	GLP shows anti-tumoral effects and can play an important role in downregulating inflammation and blood sugar. In addition, it has immunostimulatory effects and inhibits ROS formation, reducing oxidative DNA damage [37,38,39,40]. In colorectal cancer CRC, GLP shown pro-apoptotic activity on the HCT116 cell line, increasing caspase-3, caspase-8, and Fas activities [41]. Moreover, GLP is able to reactivate mutant p53 (as seen in HT-29 and SW480 colorectal cell lines) [42].	It is possible to be dosed up with 5-FU [42].
GLT (triterpene extract)	//	It has shown suppression activity on colon cancer carcinoma cells (HT-29), and also inhibited colon cancer growth in a xenograft model. Its activity is related to the ability to arrest cell cycles in the G0/G1 phase and to induce apoptosis. GLT allows the formation of autophagous vacuoles and increases the expression of some proteins, like Beclin-1 and LC-3. Autophagy is facilitated by p38 MAPK inhibition [43].	
D9-THC (and other)	*Cannabis sativa*	It may induce apoptosis in tumoral cells, inhibiting proliferation and angiogenesis. Furthermore, it inhibits cancer cell migration, allowing anti-metastatic effects [44,45,46,47,48]. D9-THC has been shown in vitro some anticancer effects on different tumoral diseases, such as breast cancer, epithelioma, glioma, lung cancer, lymphoma, neuroblastoma, pancreatic carcinoma, prostate carcinoma, skin cancer, thyroid epithelioma, and uterine-carcinoma [49].	
Cannabidiol (CBD)	//	It has shown anti-growth effects on two tumoral cell lines (HCT116 and DLD-1), however, cannabidiol derivatives are ineffective against healthy cells’ proliferation [50]. In vivo, CBD extract proved effective in reducing pre-neoplastic lesions and polyps (induced through azoxymethane) as well as in xenograft models [50]. In addition, CBD displayed chemopreventive effect on HCT116 and Caco-2 cell lines, protecting them from oxidative damage and decreased cell proliferation through CB1, TRPV1, and PPARg [47].	
Flavonoids	Several plant organs and compounds.	Flavonoids are able to promote a protective effect on cells against cancer evolution in several ways [13].	
Epigallocatechin (EGCG)	*Camellia Sinensis*	It acts on cell signalling. In fact, it can stop both the growth and migration of CRC cells. These effects are due to the inhibition of the TF/VIIa/PAR2 signalling pathway, which is very important for inducing ERK1/2 phosphorylation and activation of NF-kB. In particular, EGCG reduces NF-*k*B transcription factor activity, induces increased expression of caspase-7, and decreased expression of MMP-9 [51]. Moreover, epigenetic process could be regulated by EGCG, it allows the ubiquitination of colorectal cells sensitive to methylation, helping in DNMT3A (DNA methyltransferase 3A) and HDAC (histone deacetylases) degradation [52,53]	EGCG showed synergistic effect with 5-FU [54].
Genistein	soy, beans, lentils, and chickpeas	In the HT-29 cell line, genistein up-regulates the expression of Bax, p21 proteins and glutathione peroxidase expression. However, it can also inhibit some molecules like NF-kβ, topoisomerase II and MMP2 (this last function helps to prevent mestastases in CRC patients) [55,56]	Genistein showed synergistic effects with [57]: - 5-FU - Cisplatin For in vivo studies mice were injected 50 mg genistein/kg body weight.
Apigenin	Several plants species	Apigenin induces colorectal cell growth inhibition, decreases angiogenesis, promotes cell arrest and apoptosis in vitro [58]. In both in vitro and in vivo studies the expression of NAG-1, P53 and p21 (cell cycle inhibitor) are increased by apigenin, reducing intestinal tumour load and number [59]. Apigenin is able to inhibit ABC receptors that increase the efflux of a chemotherapeutic agent in cancer cells, on the other hand it increases their bioavailability [55].	Apigenin showed synergistic effects with Irinotecan [60].
Chrysin	honey, propolis, chamomile and martyrs (*Passiflora caerulea)*	Chrysin has shown, in HCT116, DLD-1 and SW837 cells, the ability to induce cell apoptosis. This compound increases TNFα and TNFβ genes, activating the TNF and AHR (aryl hydrocarbon receptor) signalling pathways [61]. Moreover, chrysin can induce a cell cycle arrest at the G2/M transition phase, as seen in a study on the colorectal cell line SW480 [62].	Chrysin can reduce cisplatin side effects [56]. Furthermore, it has shown a synergistic effect with Apigenin [62].
Isoliquiritigenin & Formotenin	*Glycyrrhiza Glabra* (Isoliquiritigenin) & Some plants	They inhibited growth of cancer cells lines, promoting also apoptosis [63].	Isoliquiritigenin has shown a synergistic effect with cisplatin [64].
Kaempferol	Propolis and several plants	Kaempferol increases chromatin condensation and DNA fragmentation. It is also able to increase the cleavage of caspase-9, caspase-3, and caspase-7. Together, this is how this compound induces apoptosis [65].	
Quercetin	Many types of vegetables and fruits.	Quercetin is an antiproliferative compound. It prevents the activation of RAS and inhibits migration and invasion [66,67].	
Artesunate	*Artemisia annua*	Artesunate displays anti-proliferative activity, for these reasons it is a potential anticancer agent. It is cytotoxic, allowing it to induce cell cycle arrest in the G1 phase. Furthermore, Artesunate has shown the ability to reverse immunosuppression in the cancer microenvironment [68].	Artesunate has shown synergistic effects with oxaliplatin [69].
Ginsenosides	*Panax ginseng, Panax notoginseng*	Ginsenosides decrease adhesion, inhibit migration, and cause apoptosis [70,71]	Gisenosides have shown synergistic effects with 5-FU [72].
Betulinic acid	*Betula pubescens, Pseudocydonia sinensi, Prunella vulgaris, Piteroporus betulinus, Innonotus obliquus.*	Betulinic acid targets the apoptotic mitochondrial pathway [73].	Betulinic has shown synergistic effects with [74]: - 5-FU - Oxaliplatin
Gossypol	*Gossypium*	In tumour cells, Gossypol can inhibit proliferation, inducing apoptosis [75]. It was tested on CRC and prostate cancer [75].	Gossypol has shown synergistic effects with 5-FU [75].
Isothiocyanates and indoles	*Cruciferous vegetables*	It has an important antitumoral effect, blocking cytochromes P450, and also inducing the phase II detoxification enzymes glutathione S-transferases, and promotes the elimination of carcinogens from the organism [76]. Cell metastasis and its tumoral properties (migratory and invasive mechanisms) can be inhibited by allyl isothiocyanate [77].	
Allysulfates	Alliaceae	They suppress proliferation and induce apoptosis by increasing the production of ROS in cancer cells. They also inhibit growth [78,79].	
Ethanol extract of *Aaptos suberiotides*	*Aaptos suberiotides*	The ethanol extract of *Aaptos suberiotides* inhibits cell proliferation and migration in HER2-Positive breast cancer [80].	It can reduce resistance to Trastuzumab [80].
β-Caryophyllene	Many plants	BCPO suppresses PC-3 (a prostate cancer cell line) and MCF-7 (a breast cancer cell line) growth in a dose-dependent way. Furthermore, it promotes ROS production, triggers MAPK and inhibits the PI3K/AKT/mTOR/S6K1 signalling pathway [81].	
Ethanol Extract of Marine Sponge *Stylissa carteri*	*Stylissa carteri*	This extract inhibits growth and migration, and also induces apoptosis in breast tumour cells [82].	It has shown a synergistic effect with [82]: - Doxorubricin - Paclitaxel
Cecropins	Many insect sp.	Cecropins show cytotoxic activity only against cancer cells (leukaemia, colon carcinoma, stomach, small cell lung and ovarian cancer with a multi-drug resistant phenotype) [83].	
Mycalamide A/B and Onamide extracted from sponges	*Mycale* sp. and *Theonella* sp.	These inhibit the cell-free translation of RNA in cancer cells (leukaemia) [84].	
Emericellipsin A	Extremophilic Fungus and *Emericellopsis alkalina*	Emericellipsin induces apoptosis in cancer cells (hepG2 and HeLa) [85].	
PSK	*Coriolus vescicolor*	PSK inhibits the growth of various types of tumours (fibrosarcoma, colon adenocarcinoma). Data suggest that PSK-induced immunity is tumour-specific and that T lymphocytes play an important role in antitumor memory functions [86].	It has shown a synergistic function with 5-FU [86].
AraC	*Cryptotethya crypta*	It induces apoptosis through many mechanisms. AraC, and its different metabolites, contribute to its cytotoxicity, the including incorporation of AraCTP into DNA and AraUMP into RNA, inhibition of polymerase *α* and *β*, and the impairment of repair mechanisms [87]. Besides this DNA synthesis impairment, AraC causes several signalling events, the including activation of PKC and MAPK [88] and the upregulation of AP-1 and NF-*κ*B [89,90,91].	It has shown a synergistic effect with [92]: - Idarubicin - Daunorubicin Ara-C was tested at a concentration of 10 micrograms/mL
didemin B	*Trididemnum solidum* and *Tistrella mobilis*	didemin B is cytotoxic, in vitro, for cancer cells (A549 for lung cancer and HT-29 for colorectal cancer) [93].	
bryostatins	*Bugula Neritina* and other sp.	Bryostatins cause the down-regulation of PKCs, which are translocated to the membrane and then degraded by a proteasome [94].	It has shown a synergistic effect with [94]: - AraC; - Taxol; - Tamoxifen; - Staurosporin; - Dolastatin 10; - Auristatin PE; - Vincristine; - 2-CdA; - Vincristine-AraC Phase II dose of bryostatin suggested 50 μg/m^2^/24 h [95].
cantharidin	*Blister beetles*	It causes apoptosis by a p53-dependent mechanism in leukaemia cells. Cantharidin induces both DNA single- and double-strand breaks [96].	It has shown synergy with Tamoxifen [97].
solenopsins	*Solenopsins invicta*	solenopsins inhibit PKCs’ pathway and angiogenesis [28].	

**Table 2 ijms-22-10380-t002:** Natural compounds currently used in cancer treatment and described in the chapter “Natural Compounds and Sinergy”.

Natural Product	Synergy
Ascorbic acid	Some studies have demonstrated the synergistic effects of ascorbate with other classic anti-cancer drugs.
Curcumin	Phase II clinical studies have suggested that a combination of gemcitabine and curcumin is a conceivable treatment for pancreatic cancer patients.
Epigallocatechin-3-gallate	EGCG possess synergistic effects with other anticancer drugs, such aschrysin, curcumin, erlotinib, etoposide, 5-fluorouracil, tamoxifen and temozolomide.
Resveratrol	Resveratrol exhibits synergistic effects with anti-cancer drugs, such as doxorubicin, cisplatin and vinorelbine.
Capsaicin	Some studies have suggested combinational use of capsaicin with other chemotherapeutics drugs or dietary molecules.

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
