# Peer review of "Cancer Therapy Challenge: It Is Time to Look in the “St. Patrick’s Well” of the Nature"

_ijms, 2021, doi:10.3390/ijms221910380_

Round 1
Reviewer 1 Report
Bonsignore et al. reviewed the use of compounds isolated from natural sources to fight cancer.
First, they introduced the worldwide importance of cancer in consideration of the number of deaths it causes and the trend towards an increase in the incidence. Then, they wrote a compendium on the most important natural compounds used successfully to treat cancer. They emphasized the importance of the synergistic effects brought about by the multitherapy treatment. An approach that limits the cancer's resistance to therapy and reduces side effects due to lower doses and multiple targets. Finally, they detailed available results on the possible use of four novel natural compounds for cancer treatment.
The manuscript is well organized and easy to read. However, in my opinion, the manuscript could be improved with the addition of a second table listing the natural compounds currently used in cancer treatment and described in the chapter "Natural Compounds for Cancer Therapy".
Furthermore, it would be useful to increase the number of new natural compounds detailed in the text beyond ascorbic acid, curcumin, EGCG, etc. For example the authors did not mention the use of capsaicin which might deserve a small chapter.
Author Response
Bonsignore et al. reviewed the use of compounds isolated from natural sources to fight cancer.
First, they introduced the worldwide importance of cancer in consideration of the number of deaths it causes and the trend towards an increase in the incidence. Then, they wrote a compendium on the most important natural compounds used successfully to treat cancer. They emphasized the importance of the synergistic effects brought about by the multitherapy treatment. An approach that limits the cancer's resistance to therapy and reduces side effects due to lower doses and multiple targets. Finally, they detailed available results on the possible use of four novel natural compounds for cancer treatment.
We thank the reviewer for the positive evaluation of our ms.
The manuscript is well organized and easy to read. However, in my opinion, the manuscript could be improved with the addition of a second table listing the natural compounds currently used in cancer treatment and described in the chapter "Natural Compounds for Cancer Therapy".
We have inserted Table 2, listing the natural compounds described in the chapter "Natural Compounds and Sinergy".
Furthermore, it would be useful to increase the number of new natural compounds detailed in the text beyond ascorbic acid, curcumin, EGCG, etc. For example, the authors did not mention the use of capsaicin which might deserve a small chapter.
We have followed the suggestion of the reviewer 1 and we have inserted a small chapter (4.6) about capsaicin.
Reviewer 2 Report
In the present review, the authors try to resume actual knowledge about therapeutic application of natural compounds. Unfortunately, the review of literature is poor and is not comprehensive. The authors did not present any interesting point of view or original research hypothesis. The final conclusions are rather obvious. The text is very general and with limited novelty.
It would be better if the authors try to focus on side-effects of natural compounds and limitations of using natural compounds in therapeutic application/medicine. The section about bioavailability in mammalian systems and biotransformation of natural compounds in cells is strongly recommend. Please add. In table 1, the concentrations of natural compound used in various researches should be given.
The authors completely omitted information about the effect of natural compounds on epigenetic parameters and/or miRNA profile. What about using nanocarriers as delivery systems of natural compounds?
Author Response
In the present review, the authors try to resume actual knowledge about therapeutic application of natural compounds. Unfortunately, the review of literature is poor and is not comprehensive. The authors did not present any interesting point of view or original research hypothesis. The final conclusions are rather obvious. The text is very general and with limited novelty.
We thank the reviewer for the evaluation of our ms. We have modified the manuscript to improve quality and novelty.
It would be better if the authors try to focus on side-effects of natural compounds and limitations of using natural compounds in therapeutic application/medicine. The section about bioavailability in mammalian systems and biotransformation of natural compounds in cells is strongly recommend. Please add. In table 1, the concentrations of natural compound used in various researches should be given.
We have expanded the discussion about natural compounds bioavailability. We have indicated more details about natural products concentrations in Table 1.
The authors completely omitted information about the effect of natural compounds on epigenetic parameters and/or miRNA profile. What about using nanocarriers as delivery systems of natural compounds?
We have inserted a new chapter (Natural compounds as epigenetic modulators) and we have discussed the nano-based drug delivery.
Reviewer 3 Report
This paper presents a good literature review of the natural compounds utilization in cancer therapy, however this article requires some revisions that need to be addressed prior to the publication, mostly regarding minor editing of English language and style that need improving.
- I think that the <<A huge number of drugs, such as antibody-based drugs or specific inhibitors of specific targets, have been developed following the concept “one-drug-one target”>> should be changed into <<An important number of drugs, such as antibody-based drugs or particular inhibitors of specific targets, have been developed following the concept “one-drug-one target”>>(Line 107, Page 3).
- I would change “classic anti-cancer drugs” to “common anti-cancer drugs” (Line 146, Table 1, Row 1, Page 4)
- I would rather substitute the word <<and>> from “that and other lymphomas [18] and lung tumour [19,20] and neuroblastoma [21]” with a comma, obtaining “that and other lymphomas [18], lung tumors [19,20] and neuroblastoma” (Line 146, Table 1, Row 2, Page 4).
- The typo should be corrected “brest” with “breast” (Line 146, Table 1, Row 3, Page 4).
- I would change “to administrate” to “to be dosed up with” (Line 146, Table 1, Row 4, Page 5).
- I would replace “they” with “cannabidiol derivatives” (Line 146, Table 1, Row 2, Page 6)
- In the sentence “They inhibited growth and promoted apoptosis [61].” I would elaborate a little bit more…growth of tumoral cells or of all cells…it was tested on a certain cell line (Line 146, Table 1, Row 5, Page 7).
- I would recommend removing “if” from “reduction of tumor incidence if compared to controls” (Line 288, Page 13).
- The typo should be corrected “rug” with “drug” (Line 360, Page 15).
- I would rephrase Line 360-364 there are five mentions of the word drug… (Page 15)
Author Response
We thank reviewer for taking time to comment our ms and for the positive evaluation.
We have modified the text according to suggestions.
Round 2
Reviewer 2 Report
The authors have still not focused on side-effects of natural compounds and limitations of using natural compounds in therapeutic application/medicine. The section about bioavailability in human system and biotransformation of natural compounds in human cells is still lacking. Therefore I have not recommend this manuscript for publication.